# RSMerge: Bridging Head and Tail Classes via Subsampled Model Merging

## Abstract

Class imbalance is a pervasive challenge in machine learning, where head classes have abundant samples, while tail classes are severely underrepresented. This imbalance significantly impacts predictive performance, particularly in scenarios where maintaining balanced accuracy is critical. Traditional fine-tuning methods for foundational models such as CLIP often prioritize head-class accuracy but distort pre-trained representations for tail classes, leading to suboptimal overall performance. Conversely, parameter-efficient fine-tuning (PEFT) methods preserve tail-class features but struggle to fully leverage head-class information. In this study, we first show empirically how different head-to-tail class ratios affect model performance, highlighting the limitations of existing fine-tuning methods across various imbalance distributions. To address these limitations, we propose a two-stage learning framework that merges models fine-tuned on balanced subsets via full-rank updates and then freezes the encoder to retrain the classifier on the full dataset. Validated across five benchmark datasets with distinct imbalance patterns, our method achieves superior trade-offs between head and tail class accuracies while maintaining generalizability.

## 1. Introduction

In machine learning, the assumption of balanced class distributions is deeply ingrained in both theory and practice (Deng et al., 2009; Zhou et al., 2018; Krizhevsky et al., 2009). However, real-world datasets often deviate significantly from this assumption, with head classes (frequent categories) dominating the data while tail classes (rare categories) are severely underrepresented (Van Horn et al., 2018; Holste et al., 2022; Liu et al., 2019). This imbalance poses

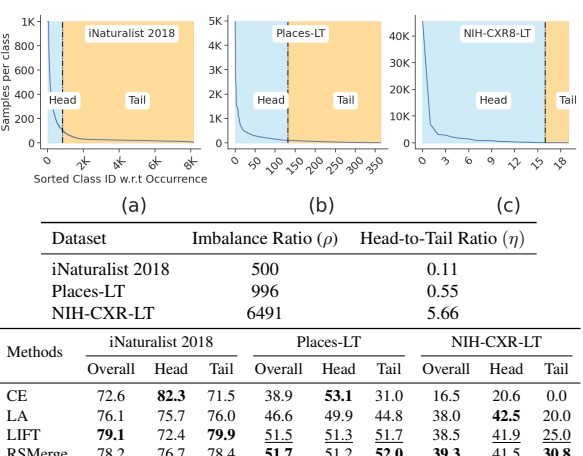

*Figure 1.* Impact of $\rho$ (degree of class skew) and $\eta$ (proportion of head vs. tail classes) on class distributions. (a–c) Example distributions with varying $\rho$ and $\eta$. While $\rho$ is widely studied, $\eta$ is a critical but often overlooked factor that determines the sensitivity of the predictor to head or tail classes. Our method achieves robust performance across all regimes demonstrating insensitivity to distributional asymmetry.

a fundamental challenge: models must effectively learn from limited tail-class samples while maintaining robust performance across all classes (Chen et al., 2024).

Traditional approaches to class imbalance, such as re-sampling (Liu et al., 2008; Kim et al., 2020; Shi et al., 2023) and re-weighting (Menon et al., 2021; Ren et al., 2020; Cao et al., 2019), aim to mitigate the effects of skewed class distributions, often called as imbalance ratio ($\rho$). While these methods can improve tail-class recognition, they often fail to address the underlying representation learning challenges, leading to suboptimal performance across the entire dataset (Wang et al., 2021). Recent advances in vision-language foundation models, particularly CLIP (Radford et al., 2021), have opened new avenues for tackling this problem. Pretrained on vast and diverse datasets, CLIP exhibits inherent robustness to class imbalance, making it a promising candidate for fine-tuning in imbalanced settings (Wen et al., 2024; Wang et al., 2023; Ma et al., 2021; Tian et al., 2022; Long et al., 2022). Notably, LIFT (Shi et al., 2024) achieves state-of-the-art results by employing parameter-efficient fine-tuning (PEFT) methods (Han et al.,

[1]Anonymous Institution, Anonymous City, Anonymous Region, Anonymous Country. Correspondence to: Anonymous Author <anon.email@domain.com>.

2024) on CLIP's vision encoder.

Despite these advances, a critical limitation remains: Existing methods often fail to adapt to varying head-to-tail ratios ($\eta$), which represent the relative frequency gap between dominant and rare classes. For instance, while LIFT excels in tail-dominated distributions (Figure 1a), its reliance on low-rank adaptation compromises performance in balanced (Figure 1b) or head-heavy scenarios (Figure 1c). This highlights a fundamental trade-off: full-rank fine-tuning enhances adaptability to specific imbalance patterns but risks catastrophic forgetting of pretrained features, while low-rank adaptation preserves generalizability at the cost of task-specific optimization.

To address this challenge, we propose a two-stage framework that strategically balances adaptability and stability. In the first stage, we progressively subsample the training data to create balanced subsets, fine-tune independent CLIP encoders on each subset via full-rank updates, and merge their parameters to retain pretrained feature robustness. In the second stage, we recycle all discarded data by freezing the merged encoder and retraining only the classification head on the full dataset. This approach ensures robust performance across diverse imbalance patterns without distorting learned representations.

Our work makes three key contributions: **(1)** We formally characterize the impact of head-to-tail ratios on model performance, providing a comprehensive analysis of different imbalance manifestations. **(2)** Our two-stage approach reconciles the stability-plasticity dilemma through decoupled learning and model merging, achieving better trade-offs between head and tail class accuracies across diverse scenarios. **(3)** We validate our approach on five benchmark datasets, each representing distinct imbalance distributions, demonstrating its effectiveness and generalizability in real-world applications.

## 2. Related Work

### 2.1. Imbalanced Classification

We can roughly divide progress on imbalanced classification into three groups.

**Re-sampling/Re-weighting.** Class imbalance mitigation strategies broadly involve oversampling minority classes (Chawla et al., 2002), subsampling majority classes (Liu et al., 2008), or reweighting losses (He & Garcia, 2009). Subsampling risks losing majority-class discriminative patterns, oversampling may overfit minority classes (Zhou et al., 2020), and reweighting struggles in overparameterized networks (Zhai et al., 2023). Recent advances like *logit adjustment* loss (LA) (Menon et al., 2021; Ren et al., 2020) addresses these issues by enforcing larger margins for tail

classes, bridging data imbalance with geometric regularization. However, in our work, we argue LA loss is insufficient to address full fine-tuning from foundational models.

**Decoupled Learning.** Decoupled learning frameworks address class imbalance through sequential training phases: representation learning via instance-balanced sampling followed by classifier refinement using class-balanced strategies (Kang et al., 2020; Zhang et al., 2021). This paradigm assumes model biases primarily reside in the classifier layer, positing that head-tail performance gaps can be resolved through post-hoc classifier calibration (Izmailov et al., 2022; Yang et al., 2023). However, we demonstrate this assumption becomes invalid when fine-tuning foundational models – neglecting tailored strategy for representation learning degrades both head and tail class performance due to catastrophic forgetting of pre-trained features (Shi et al., 2024; Mukhoti et al., 2023).

**Ensemble Learning.** Ensemble methods address data imbalance by combining specialized experts trained on complementary distributions (Cai et al., 2021; Li et al., 2022). Notable approaches include: BBN's dual-branch architecture balancing original and re-sampled distributions (Zhou et al., 2020); RIDE's dynamic routing of instances to distribution-aware experts (Wang et al., 2021); and LFME's multi-teacher distillation across many/medium/few-shot groups (Xiang et al., 2020). While effective, these methods rely on heuristic expert specialization rules and often result in cumbersome architectures that hinder adaptation to foundational models, increase training complexity, and limit inference speed. Our work circumvents these limitations through two key innovations: (1) replacing specialized expert design with parallel fine-tuning of foundation models on controlled subsamples, and (2) employing model averaging and EMA instead of complex aggregation mechanisms. This preserves the ensemble's variance-reduction benefits while maintaining the architectural simplicity and computational efficiency of the original foundation model.

### 2.2. Model Merging

Model merging, also sometimes refer to weight averaging, has gained significant attention in recent years as a promising research direction (Li et al., 2023), focusing on reducing communication costs in federated learning (McMahan et al., 2017) and distributed training (Douillard et al., 2023), enabling the efficient combination of multiple models without additional training (Ilharco et al., 2023), and enhancing model robustness in out-of-distribution scenarios (Wortsman et al., 2022a; Rame et al., 2022). Early approaches like Exponential Moving Average (EMA) (Tarvainen & Valpola, 2017) and Stochastic Weight Averaging (SWA) (Izmailov et al., 2018) have been widely adopted to accelerate train-

ing convergence and enhance the generalization capabilities of deep neural networks. Recent work extends merging to sequential adaptation: Alexandrov et al. (Alexandrov et al., 2024) mitigate catastrophic forgetting in continual pretraining via iterative merging, while Ramé et al. (Ramé et al., 2024) align LLMs through multi-stage averaging during RLHF. To our knowledge, no prior work applies model merging to imbalanced recognition. Unlike existing sequential merging approaches, our framework trains multiple models in parallel on complementary subsampled distributions – a critical design choice for handling long-tailed data. We propose the first schema specifically tailored for imbalance, integrating subsampling (to retain tail-class discriminability) and resampling (to stabilize head-class representations). This parallelized merging strategy directly addresses feature-space asymmetry in long-tailed distributions while maintaining computational efficiency, enabling foundational models to adapt to extreme imbalance without sacrificing pre-trained generalization.

### 2.3. Imbalanced Learning with Foundational Models

Foundation models such as CLIP exhibit inherent robustness to class imbalance, as demonstrated by their zero-shot generalization capabilities (Wen et al., 2024). Recent advances further enhance this property through retrieval-augmented architectures (Tian et al., 2022; Long et al., 2022), prompt-tuning strategies (Dong et al., 2022; Xia et al., 2023), and joint vision-language training paradigms (Ma et al., 2021; Wang et al., 2023). While these methods improve adaptation to long-tailed distributions, LIFT (Shi et al., 2024) reveals that PEFT with LA loss achieves state-of-the-art performance by selectively adapting CLIP's pre-trained features. We demonstrate that LIFT tail-class gains come at the cost of degraded head-class accuracy—a critical flaw in applications requiring balanced performance. Our framework reduces this compromise through regularizing weight updates via averaging across complementary subsampled distributions, and full-rank optimization.

## 3. Imbalanced Learning with Foundational Models

### 3.1. Preliminaries

Given training data $D = \{\boldsymbol{x}_i, y_i\}_{i=1}^N$, where $\boldsymbol{x}_i$ is a training sample and $y_i \in \mathcal{C}$ is class label with cardinality of $K$. We assume that training data follow an imbalanced class distribution where the class prior distribution $\mathbb{P}(y)$ is highly skewed so that there exist some underrepresented classes with a very low probability of occurrence. Specifically, we define the imbalance ratio as $\rho = \max_y \mathbb{P}(y) / \min_y \mathbb{P}(y)$ to indicate the skewness of data. Classes with high $\mathbb{P}(y)$ are referred to as *head classes*, while others are referred

to as *tail classes*. We define head classes as those with over 100 training samples[1] (Liu et al., 2019). At test time, as we are interested in obtaining a predictor capable of recognizing all classes well, we maximize the BalAcc = $\frac{1}{|\mathcal{C}|} \sum_{c \in \mathcal{C}}$ Accuracy$(c)$. We decompose the model into a feature extraction and a classification head components. For feature extraction, we use CLIP's vision encoder implemented by a ViT (Dosovitskiy et al., 2021), parameterized by $\theta$, defined as $f_I(\boldsymbol{x}; \theta) = \boldsymbol{z}$, where $\boldsymbol{z}$ represents the extracted feature for input $\boldsymbol{x}$. The final class prediction $\hat{y} = \arg\max g(\boldsymbol{z}; w)$ is produced by a classification head $g$ with parameters $w$. Following (Radford et al., 2021), we adopt a prototypical classification head for $g$, where both features and classifier weights are $l_2$-normalized, and a temperature is applied to the logits. The parameters $w$ are initialized by generating textual prompts[2] for the classes and extracting corresponding textual features using the CLIP text encoder (Shi et al., 2024).

During training, typically guided by the Empirical Risk Minimization (ERM) framework, the cross-entropy loss is minimized as follows:

$$\ell(y, g(\boldsymbol{z})) = -\log \frac{\exp(g_y(\boldsymbol{x}))}{\sum_{y' \in \mathcal{C}} \exp(g_{y'}(\boldsymbol{x}))} \quad (1)$$

where $g_y$ denotes the predictive logit of model on class $y$. However, this ubiquitous approach neglects the issue of class imbalance and makes the model biased toward head classes. *Logit Adjustment* (LA) (Menon et al., 2021) loss instead corrects head class biases by adding a label-dependent offset to each of the logits as follows:

$$\ell_{LA}(y, g(\boldsymbol{z})) = -\log \frac{\exp(g_y(\boldsymbol{x}) + \log \pi_y)}{\sum_{y' \in \mathcal{C}} \exp(g_{y'}(\boldsymbol{x}) + \log \pi_{y'})} \quad (2)$$

where $\pi \in \Delta_y$ are estimates of the class priors $\mathbb{P}(y)$ based on the empirical class frequencies on the training data $D$.

In a recent study, Shi et al. (2024) observed that starting from CLIP pre-trained weights, the LA loss alone is insufficient to achieve strong performance. Specifically, they showed that full fine-tuning with LA, referred to as Full-FT hereafter, often results in inconsistent class-conditional distributions, particularly among tail classes. To address this, they suggest maintaining proximity to the pre-trained initialization by leveraging PEFT methods.

### 3.2. Head-Tail Trade-off for Imbalance Recognition

In practice, imbalanced distributions can manifest in various forms. A crucial but often overlooked factor is the frequency

---

[1]Here, we define a single subset as the tail to simplify the analysis. Later, in the experiment section, we further divide the tail into two smaller subsets—med-shots and few-shots—for consistency with existing works.

[2]We use descriptive prompts such as "a photo of a cat" or "a photo of a dog" to represent each class (Radford et al., 2021).

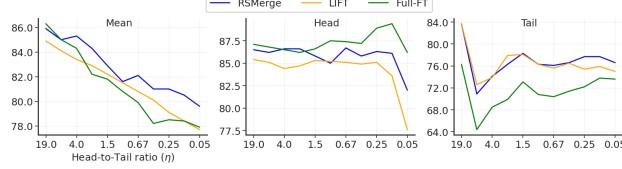

Figure 2. Performance vs. head-tail ratio on proposed CIFAR100-LT benchmark. Our proposed approach obtains good accuracy for different levels of $\eta$ for both head and tail classes

ratio between head and tail classes, which directly influences predictive performance.

**Definition 3.1.** Let $\mathcal{H} = \{c \mid n_c > \tau\}$ and $\mathcal{T} = \{c \mid n_c \leq \tau\}$ denote the partitioning of classes based on a sample threshold $\tau$ for head and tail classes, respectively. Define $H = |\mathcal{H}|$ and $T = |\mathcal{T}|$ as the number of head and tail classes. The head-to-tail ratio is given by $\eta = \frac{H}{T}$.

While the LA loss is effective in correcting skewed class priors, it does not address discrepancies in the head-to-tail ratio. When the training distribution has a high $\eta$ (i.e., many head classes relative to tail classes), models benefit from focusing on head classes to achieve high accuracy. Conversely, when $\eta << 1$, prioritizing tail classes becomes sufficient for good performance. To better understand the influence of $\eta$ on the final performance, we design a synthetic dataset based on CIFAR-100 with a fixed imbalance ratio $\rho = 100$ but varying $\eta$. Specifically, for a given $\eta$, we partition the classes into head and tail subsets. Within each subset, class sample sizes follow an exponential decay distribution, ranging from 500 to 101 for head classes and from 100 to 5 for tail classes. This process is repeated for $\eta$ values ranging from 19 to 0.05 (see appendix for visualization of resulting imbalance distributions (Figure 5)). As shown in Figure 2, our results confirm that full fine-tuning consistently outperforms LIFT for head classes, while LIFT excels in tail classes. As the proportion of head classes decreases and tail classes dominate, the mean accuracy shifts in favor of LIFT, supporting our hypothesis. Notably, these findings align with recent observations in LLM literature, where LIFT often underperforms full fine-tuning but better preserves the base model's performance on tasks outside the target domain (Biderman et al., 2024).

## 4. RSMerge: Imbalanced Learning by Controlling Weight Change

Our findings uncover a fundamental trade-off in model optimization: full fine-tuning, which updates the full-rank weight matrices, excels in head-class generalization but significantly compromises tail-class accuracy. Conversely, LoRA enhances tail-class performance by maintaining weights close to the pre-trained initialization, yet it sacri-

**Algorithm 1** RSMerge (Parallelizable Pseudocode)
1: **Input:** zero-shot weights $\theta_0$, $N$ imbalance ratios $\{\rho_n\}$, $M$ runs per $\rho_n$, $T$ training steps, $\mu$ EMA rate, optimizer Opt, training data $D$
2: $\{\bar{\theta}^n\}_{n=1}^N \leftarrow \emptyset$
   *Parallel section (runs $n = 1..N$):*
3: **for** $n = 1$ to $N$ **do**
4:     $\{\theta_{ema}^m\}_{m=1}^M \leftarrow \emptyset$
       *Parallel section (runs $m = 1..M$):*
5:     **for** $m = 1$ to $M$ **do**
6:         Subsample $D$ via $\rho_n$ to create $D_m$
7:         Initialize $\theta^m, \theta_{ema}^m \leftarrow \theta_0$
8:         **for** $t = 1$ to $T$ **do**
9:             Sample $(x, y) \sim D_m$
10:             $\theta^m \leftarrow \text{Opt}(\theta^m, \nabla_\theta[\ell_{LA}(y, f(x))])$
11:             $\theta_{ema}^m \leftarrow (1 - \mu)\theta_{ema}^m + \mu\theta^m$
12:         **end for**
13:         Save $\theta_{ema}^m$
14:     **end for**
15:     $\bar{\theta}^n \leftarrow \frac{1}{M} \sum_{m=1}^M \theta_{ema}^m$
16:     Save $\bar{\theta}^n$
17: **end for**
18: Calculate $\Theta_N$ from Equation (5)
19: Re-train final classifier on full $D$

fices head-class accuracy. Striking a balance between these objectives necessitates careful control over weight updates while preserving the benefits of full-rank optimization.

To tackle this challenge, we introduce *Resample/Subsample Model Merging (RSMerge)*, a novel model merging framework tailored for imbalanced recognition tasks. *RSMerge* operates in two distinct phases, as illustrated in Figure 3 and outlined in Algorithm 1. **(1)** In the representation learning phase, we train multiple models in parallel on resampled versions of the training distribution, followed by a model merging step. This strategy ensures proximity to the pre-trained weights while effectively capturing essential features from head classes. **(2)** In the second phase, we leverage the previously discarded data from the first stage by fine-tuning only the classifier head using the LA loss on the full dataset. The remainder of this section provides a comprehensive description of our method, along with experimental validation of our design choices and their impact on performance.

### 4.1. Representation Learning

The decoupled learning framework has emerged as a powerful approach for addressing class imbalance, achieving state-of-the-art performance across numerous benchmarks (Yang et al., 2023). Conventional methods often assert that correcting the classifier head alone is sufficient to handle imbalanced data distributions (Kang et al., 2020). These approaches typically rely on instance-balanced sampling

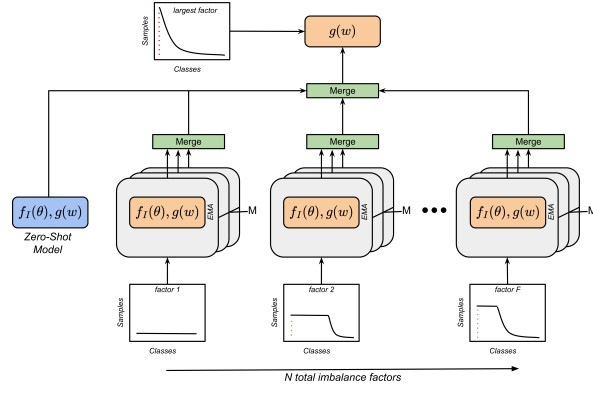

*Figure 3.* RSMerge diagram: (1) Merging models fine-tuned on $M$ balanced subsets and $N$ progressively varying imbalance ratios to preserve pretrained features while adapting to class distribution shifts. (2) Freezing the merged encoder and retraining the classifier on the full dataset to leverage all training samples without representation distortion.

paired with vanilla cross-entropy loss for representation learning, followed by reweighting or resampling techniques to refine the classifier (Zhang et al., 2021).

However, with the advent of foundational models like CLIP, neglecting the representation learning component can have significant drawbacks (Shi et al., 2024; Wang et al., 2023). Notably, CLIP's zero-shot performance on many benchmark datasets demonstrates remarkable balance across both head and tail classes (Wen et al., 2024). Careless fine-tuning of these representations can degrade the pre-trained features (Mukhoti et al., 2024), which are challenging to restore during subsequent classifier training. To address this challenge, we propose four strategies designed to preserve the valuable pre-trained knowledge while enabling the model to extract task-relevant features for downstream tasks.

**Exponential Moving Average (EMA).**  Fine-tuning risks driving the model away from its pretrained state, leading to catastrophic forgetting of pretrained knowledge and representation collapse. This can harm generalization, particularly for tail classes, as CLIP's pretrained model itself is a strong zero-shot learner (Wen et al., 2024). To counteract this, *RSMerge* maintains an exponential moving average of the model parameters throughout fine-tuning, updated at each step with a momentum coefficient $\mu = 0.01$:

$$\theta_{ema} = (1 - \mu) \cdot \theta_{ema} + \mu \cdot \theta_0. \tag{3}$$

EMA averages parameters from both the initialization phase and the converged minima, acting as a regularizer that pulls the model closer to its initial state (Huang et al., 2017). Additionally, EMA encourages convergence to flatter minima (Izmailov et al., 2018), improving generalization, particu-

larly for tail classes.

**Progressive Subsampling.**  Subsampling is a common strategy to address class imbalance, typically involving the removal of data from overrepresented classes (He & Garcia, 2009). However, aggressive subsampling risks discarding valuable information from head classes, which can degrade overall model performance (Kim et al., 2020; Chawla et al., 2002; Shi et al., 2023). To mitigate this issue, we propose *progressive subsampling*, a method that incrementally increases the dataset's imbalance ratio across multiple independent rounds. While there are multiple ways to achieve this, we adopt a straightforward approach: starting with a balanced dataset, we double the imbalance ratio in each round for a total of $N$ rounds (see the appendix for visualization of resulting subsampled distributions (Figure 6)).

Each model is fine-tuned independently on its respective subsampled split, employing the LA loss function to adapt to shifts in the label distribution when necessary. This progressive approach alleviates the challenges posed by extreme imbalance by preserving tail-class data through controlled subsampling (Ren et al., 2020). Additionally, it enhances worst-class generalization by restoring geometric symmetry to the classifier (Chaudhuri et al., 2023), ensuring robust performance across all classes. For all experiments, unless explicitly stated otherwise, we set $N = 6$ corresponding to imbalance ratios $\rho \in \{1, 2, 4, 8, 16, 32, 64\}$.

**Resampling of Subsamples.**  Discarding data during the subsampling step can increase the variance in fine-tuned models.  A common strategy to mitigate this variance is model ensembling (Breiman, 1996). In our approach, we leverage bootstrapping (bagging) and the inherent randomness of SGD optimization to promote model diversity. Specifically, for bootstrapping, we fine-tune $M$ independent models, each trained on a dataset sampled from an imbalanced distribution with a fixed imbalance ratio $\rho$. Additionally, we introduce further stochasticity by randomizing batch ordering and applying diverse data augmentation techniques (Lakshminarayanan et al., 2017). Unless otherwise stated, we set $M = 2$ for all experiments.

**Model Merging.**  The preceding stages produce $NM$ independently trained models, each designed to ensure balanced representation and confident predictions. In this stage, we introduce two model merging techniques to consolidate these models into a single unified model for the subsequent phase of training.

For the resampling step, consider $M$ independently fine-tuned models, all trained on the same distribution. We merge

their weights uniformly using the following formulation:

$$\bar{\theta}_n = \frac{1}{M} \sum_{m=1}^{M} \theta_n^m \qquad (4)$$

Here, $\theta_n^m$ represents the weights of the model trained on the $m$-th resample split from the $n$-th subsampled distribution. This weight-averaging process reduces variance within each model and enhances robustness (Dietterich, 2000; Lakshminarayanan et al., 2017).

For the subsampling step, given a sequence of models $\{\theta_n\}_{n=0}^{N}$ initialized with pretrained weights $\theta_0$, we recursively merge them through weighted interpolation:

$$\Theta_k = \begin{cases} \theta_0 & \text{if } k = 0, \\ \lambda\Theta_{k-1} + (1-\lambda)\theta_k & \text{for } k = 1, \dots, N \end{cases} \qquad (5)$$

where $\lambda$ controls the preservation of earlier merged knowledge. The final merged model. The final merged model $\Theta_N$ inherits two key properties. First, its proximity to $\theta_0$ ensures retention of CLIP's zero-shot capabilities (Wortsman et al., 2022b). Second, progressive averaging models in ascending order of imbalance ratio, encode complementary head/tail class features (Zhou et al., 2020).

### 4.2. Classifier Re-Training

To address partial data utilization in Stage 1, we freeze the backbone and retrain only the classifier head on the full dataset using LA loss. This preserves Stage 1 representations while recovering discarded head-class samples. LA loss encodes label frequencies into the objective, recalibrating decision boundaries to reflect the true distribution. Freezing the backbone stabilizes the feature space, forcing the classifier to adapt without distorting representations, ensuring tail classes retain discriminative power and head classes regain accuracy.

### 4.3. Empirical analysis of RSMerge

In this section, we conduct a comprehensive analysis of *RSMerge* through multiple perspectives. First, we demonstrate that *RSMerge* achieves a balance between head and tail class performance by carefully controlling weight updates, resulting in significantly lower weight magnitudes compared to full fine-tuning while surpassing LIFT in final accuracy. Second, we evaluate the confidence of *RSMerge* predictions using various calibration metrics, highlighting its improved reliability. Third, we investigate the compatibility of PEFT methods with the *RSMerge* framework, providing insights into why PEFT may not be suitable in this context. Finally, we emphasize the computational efficiency of *RSMerge*, which leverages parallel training on subsampled distributions to optimize resource utilization. All empirical analyses

are conducted on the TinyImageNet-LT dataset, a benchmark constructed by exponentially decaying the sample sizes across 200 classes, ranging from 500 samples for the most frequent classes to just 5 samples for the rarest.

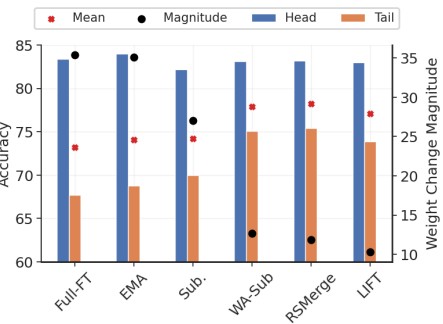

*Figure 4.* Classifiaciton accuracy for head, tail, mean, and weight change magnitude for different methods on TinyImageNet.

**Weight magnitude analysis.** A key factor behind the success of *RSMerge* is its ability to keep fine-tuned weights close to the pre-trained initialization while enabling the model to adapt to new task-specific features. To better understand how each component contributes to this balance, we analyze the impact of our method on weight change magnitude and accuracy for both head and tail classes. Starting with the baseline, we sequentially incorporate each component and track their effects.

As shown in Figure 4, full fine-tuning results in the largest weight change magnitude, while LIFT produces the smallest. The introduction of EMA improves performance for both head and tail classes while slightly reducing weight magnitude. Progressive subsampling, which trains on less imbalanced distributions, significantly limits weight changes and boosts tail-class accuracy, albeit at the cost of reduced head-class performance. Model merging, even without resampling, recovers head-class accuracy by averaging multiple models trained on different imbalanced distributions and enhances tail-class accuracy by incorporating the pre-trained model into the merging process. The final version which incorporates resampling, further refines performance by reducing noise in the trained models. Notably, *RSMerge* achieves a slightly higher weight magnitude than LIFT, which we argue contributes to its superior final performance by striking an optimal balance between adaptation and stability.

**Model calibration analysis** Table 1 An inherent advantage of model merging methods is their ability to improve prediction calibration metrics. We evaluate RSMerge against LIFT and Full-FT by measuring Negative Log-Likelihood (NLL), Expected Calibration Error (ECE) (Naeini et al., 2015), and Brier score (Brier, 1950). For NLL and Brier scores, we also provide category-wise

results. All metrics are computed after temperature tuning on the validation set. As shown in Table 1, RSMerge consistently outperforms the other methods on TinyImageNet-LT in terms of calibration.

| Method | Metric | Mean | Head | Tail |
|--------|--------|------|------|------|
| Full-FT | ECE | 1.97 | - | - |
| | Brier Score | 0.36 | 0.21 | 0.40 |
| | NLL | 1.03 | 0.63 | 1.25 |
| LIFT | ECE | 1.95 | - | - |
| | Brier Score | 0.32 | 0.23 | 0.35 |
| | NLL | 0.89 | 0.68 | 0.99 |
| RSMerge | ECE | **1.36** | - | - |
| | Brier Score | **0.30** | **0.20** | **0.33** |
| | NLL | **0.83** | **0.59** | **0.97** |

*Table 1.* Calibration metrics on TinyImageNet for Full-FT, LIFT, and our RSMerge.

**PEFT compatiblity** A natural question is whether the full fine-tuning process in RSMerge can be replaced by PEFT methods. To investigate this, we use LoRA as a representative approach. In the representation learning stage, we freeze the CLIP pre-trained weights and introduce LoRA parameters to the attention and MLP layers of the ViT, tuning their learning rate and rank. For model merging, the LoRA parameters are combined with the pre-trained weights before applying our merging schema. Finally, we retrain the classifier using the LA loss. The performance (77.1 vs 77.2) matches that of end-to-end LoRA training. We hypothesize this outcome is due to a phenomenon observed in LLM literature (Shuttleworth et al., 2024), where LoRA introduces high-ranking singular vectors (intruder dimensions) that are absent in full fine-tuning. While these models achieve comparable task performance, they adapt less robustly to sequential tasks and diverge from the pre-training distribution.

**Computational analysis** RSMerge involves a total of $NM + 1$ training runs, where $M$ models are trained on different subsampled distributions, $N$ models are trained per subsampling round by resampling the target distribution, and a final classifier is trained on the full data distribution. To evaluate the training complexity of RSMerge, two key factors are important: **(1)** RSMerge operates on severely subsampled distributions, meaning the training data in each round is much smaller than the full dataset—progressive subsampling, for example, often limits the imbalance ratio to 64, corresponding to approximately 65% of the full dataset (details in the appendix). **(2)** During the first stage (representation learning), all models are trained independently, allowing parallel execution. As a result, the overall training time is dictated by the longest single training run, typically corresponding to the subsampled distribution with the highest imbalance ratio (e.g., 64).

In terms of memory complexity, compared to full fine-tuning we also need to maintain the EMA version of the base model inside the memory.

## 5. Experiments

**Datasets and evaluation protocol.** We evaluate our method on both synthetic and real-world datasets. For synthetic cases, we use ImageNet-LT, Places-LT, and CIFAR-100-LT, which exhibit Pareto or exponential decay distributions with class sample sizes ranging from 1,280 to 5 images. For real-world scenarios, we test on iNaturalist 2018 (8,142 species, 437.5K images) and NIH-CXR-LT (20 classes, 88,637 images), representing distinct imbalance patterns with 10% and 90% head classes, respectively. Following Liu et al. (2019), we report accuracy across many-shot ($>$100 images), medium-shot (20–100 images), and few-shot ($<$20 images) subsets. We report the baseline results without test-time augmentation, as it offers an orthogonal improvement. To conserve space, only the CLIP-based method is presented in the main manuscript. Further details on results, datasets, and baselines are available in the appendix.

### 5.1. Main results

**Synthetic datasets.** We report the test accuracy in Table 2, Table 3 and Table 4 for CIFAR100-LT, Places-LT and ImageNet-LT respectively. As shown in the table, our method achieves state-of-the-art accuracy in all datasets. Unlike LIFT, which prioritizes tail classes via low-rank updates, our full-rank optimization leverages medium/many-shot samples more effectively while maintaining competitive few-shot performance. This highlights the advantage of balancing adaptation (via full-rank updates) and stability (via subsampling and merging).

**Real-world datasets.** We validate our method on iNaturalist 2018 (89% few-shot and med-shot classes) and NIH-CXR-LT (predominantly many-shot classes), representing opposing extremes of class imbalance. As shown in Table 6 for iNaturalist, while LIFT achieves higher overall accuracy by specializing in tail-class optimization, our method demonstrates balanced improvements across all tiers (many/medium/few-shot), reflecting its generalizability beyond extreme-tail regimes. For NIH-CXR-LT (Table 5) our approach excels due to two key factors: **(1)** NIH-CXR-LT's chest X-ray images exhibit significant divergence from CLIP's pretraining distribution (evidenced by lower linear probe accuracy vs. iNaturalist). **(2)** Full-rank fine-tuning enables better feature adaptation to novel medical semantics – a capability fundamentally limited by low-rank approximations in LIFT.

| Methods | Overall | Many | Medium | Few |
|---|---|---|---|---|
| **ViT-B/16 CLIP pre-trained backbone** | | | | |
| Linear Prob (LA) | 70.0 | 77.2 | 71.1 | 60.4 |
| Full-FT (LA) | 79.6 | 88.1 | 79.9 | 69.3 |
| cRT (Kang et al., 2020) | 78.8 | 89.7 | 79.7 | 65.1 |
| LIFT (Shi et al., 2024) | 81.3 | 85.2 | 80.9 | 77.1 |
| RSMerge (Ours) | **83.5** | **88.2** | **83.5** | **78.0** |

*Table 2.* Accuracy for different methods on CIFAR100-LT.

| Methods | Overall | Many | Medium | Few |
|---|---|---|---|---|
| **ViT-B/16 CLIP pre-trained backbone** | | | | |
| Linear Prob (LA) | 48.8 | 48.8 | 49.7 | 47.1 |
| cRT (Kang et al., 2020) | 44.4 | 51.0 | 43.1 | 35.4 |
| BALLAD (Ma et al., 2021) | 49.5 | 49.3 | 50.2 | 48.4 |
| Decoder (Wang et al., 2023) | 46.8 | - | - | - |
| LPT (Dong et al., 2022) | 50.1 | 49.3 | 52.3 | 46.9 |
| Full-FT (LA) | 46.6 | 49.9 | 46.3 | 41.4 |
| cRT (Kang et al., 2020) | 44.4 | 51.0 | 43.1 | 35.4 |
| LIFT (Shi et al., 2024) | **51.5** | 51.3 | 52.2 | **50.5** |
| RSMerge (Ours) | **51.7** | 51.2 | **52.8** | 50.3 |

*Table 3.* Accuracy for different methods on Places-LT.

| Methods | Overall | Many | Medium | Few |
|---|---|---|---|---|
| **ViT-B/16 CLIP pre-trained backbone** | | | | |
| Linear Prob (LA) | 60.4 | 48.9 | 60.0 | 63.9 |
| Decoder (Wang et al., 2023) | 59.2 | - | - | - |
| LPT (Dong et al., 2022) | 76.1 | - | - | 79.3 |
| Full-FT (LA) | 76.1 | 75.7 | 76.9 | 75.3 |
| LIFT (Shi et al., 2024) | **79.1** | 72.4 | **79.0** | **81.1** |
| RSMerge (Ours) | 78.2 | **76.7** | 78.5 | 78.2 |

*Table 6.* Comparison of methods for training on iNaturalist 2018. LIFT performs better because most of the classes are among few-shots and med-shots.

## 5.2. Ablations

**Merging interpolation** $(\lambda)$**.** For CIFAR100-LT, ImageNet-LT, and Place-LT, we set $\lambda = 0.7$, while for iNaturalist and NIH-CXR-LT, we use $\lambda = 0.3$. Empirically, we observe that datasets with distributions closely aligned with the pre-trained distribution of CLIP benefit from a higher $\lambda$, whereas datasets with significant distribution shifts require a smaller $\lambda$. This choice is further supported by analyzing the performance gap between **linear probing** and **full fine-tuning** on each dataset. A large gap indicates that the pre-trained representations are already well-suited for the downstream task, reducing the need for extensive feature adaptation. Conversely, a smaller gap suggests that additional feature adaptation is necessary to achieve optimal performance.

| Methods | Overall | Many | Medium | Few |
|---|---|---|---|---|
| **ViT-B/16 CLIP pre-trained backbone** | | | | |
| Linear Prob (LA) | 74.2 | 77.8 | 73.3 | 67.4 |
| BALLAD (Ma et al., 2021) | 75.7 | 79.1 | 74.5 | 69.8 |
| Decoder (Wang et al., 2023) | 73.2 | - | - | - |
| Full-FT (LA) | 73.9 | 79.8 | 71.9 | 63.9 |
| cRT (Kang et al., 2020) | 72.6 | 81.1 | 70.6 | 56.1 |
| LIFT (Shi et al., 2024) | 77.0 | 80.2 | **76.1** | **71.5** |
| RSMerge (Ours) | **77.4** | **81.2** | 76.1 | 70.7 |

*Table 4.* Accuracy for different methods on ImageNet-LT.

| Methods | Overall | Many | Medium | Few |
|---|---|---|---|---|
| **ViT-B/16 CLIP pre-trained backbone** | | | | |
| Linear Prob (LA) | 17.5 | 13.3 | 21.1 | 16.7 |
| BALLAD (Ma et al., 2021) | 34.5 | 36.7 | 38.9 | 20.8 |
| Full-FT (LA) | 38.0 | **43.8** | **41.5** | 20.0 |
| cRT (Kang et al., 2020) | 37.7 | 42.9 | 39.3 | 25.0 |
| LIFT (Shi et al., 2024) | 38.5 | 43.3 | 40.4 | 25.5 |
| RSMerge (Ours) | **39.3** | 42.4 | 40.7 | **30.8** |

*Table 5.* Accuracy for different methods and different pre-trained backbones on NIH-CXR-LT.

**Subsampling rounds** $N$**.** Table 7 shows the sampling round we use for each dataset. Empirically, we identify two key factors that influence the choice of $N$: (1) Increasing $N$ beyond half of the total imbalance ratio leads to diminishing or even negative returns. This occurs because the representation becomes overly skewed toward the head classes, making it difficult to recover the balance using models fine-tuned on less imbalanced distributions. (2) For datasets with distributions closely aligned with the pre-trained distribution of CLIP (as argued earlier), additional rounds of subsampling are beneficial. These datasets gain more from feature learning in the head classes, which is enhanced by a subsampling using a higher imbalance ratio.

## 6. Conclusion

In this work, we address the challenge of long-tailed recognition by proposing a novel two-stage framework that balances adaptation and stability. Our approach leverages full-rank optimization to effectively utilize medium- and many-shot samples while maintaining robust performance on tail classes through progressive subsampling and model merging. By systematically analyzing the impact of head-to-tail ratios, we demonstrate that existing methods, such as LIFT, often sacrifice head-class performance for tail-class gains due to their reliance on low-rank updates. In contrast, our method achieves a superior trade-off, outperforming state-of-the-art baselines across multiple benchmarks.

## Impact Statement

Our work advances the field of long-tailed recognition by improving model performance across imbalanced datasets, which are prevalent in real-world applications such as medical imaging, wildlife monitoring, and autonomous driving. By enhancing accuracy for both head and tail classes, our method promotes fairness and inclusivity in AI systems, reducing biases toward dominant categories.

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

## A. Baselines and implementation details.

We use CLIP with the ViT-B/16 backbone and optimize the model using the AdamW optimizer (Loshchilov & Hutter, 2019). The batch size is set to 128, with learning rates of $3e-4$ for both the representation and the classification stage. A cosine decay learning rate scheduler is employed, gradually reducing the learning rate to $0.1 \cdot max\_lr$ after a warmup period spanning $max(100, 0.01 \cdot total\_steps)$ steps. The validation set of each dataset is used to select the best checkpoint.

## B. Ablations

| Dataset | $N$ | Training Percentage |
|---|---|---|
| CIFAR100-LT | 5 | 67 |
| Places-LT | 5 | 63 |
| ImageNet-LT | 7 | 79 |
| iNaturalist | 8 | 90 |
| NIH-CXR-LT | 8 | 24 |

Table 7. Number of sampling rounds ($N$) and the corresponding percentage of training data used for each dataset.

## C. Visualization

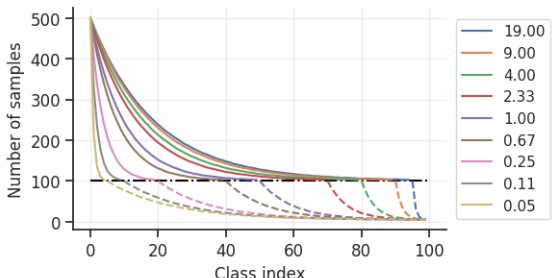

Figure 5. Visualization of imbalance distributions in CIFAR100-LT with varying values of $\eta$.

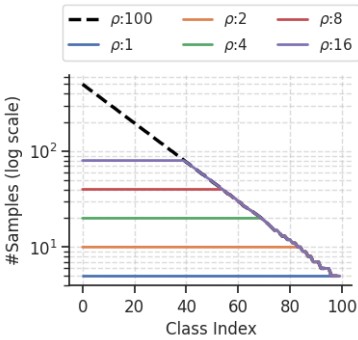

Figure 6. Example of subsampled distributions used in RSMerge, with the x-axis shown on a logarithmic scale.

## D. Dataset details.

- **ImageNet-LT** (Liu et al., 2019):
    - 115.8K images, 1,000 classes.
    - Class sample sizes: 1,280 (head) to 5 (tail).

– Constructed by Pareto sampling from ImageNet.

- **Places-LT** (Liu et al., 2019):

    – 62.5K images, 365 classes.
    – Class sample sizes: 4,980 (head) to 5 (tail).
    – Subsampled from Places365 using Pareto distribution.

- **CIFAR-100-LT** (Cao et al., 2019):

    – 100 classes, exponential decay distribution.
    – Class sample sizes: 500 (head) to 5 (tail).

- **iNaturalist 2018** (Van Horn et al., 2018):

    – 437.5K images, 8,142 species.
    – Class sample sizes: 1,000 (head) to 2 (tail).

- **NIH-CXR-LT** (Holste et al., 2022):

    – 88,637 images, 20 classes (7 head, 10 medium, 3 tail).
    – Training set: 68,058 images.
    – Test set: 20,279 images.
    – Validation/balanced test sets: 15 and 30 images per class, respectively.

# E. Full results

| Methods | Backbone | Overall | Many | Medium | Few |
|---|---|---|---|---|---|
| **Training from scratch** | | | | | |
| LDAM (Cao et al., 2019) | ResNet-32 | 42.0 | - | - | - |
| BBN (Zhou et al., 2020) | ResNet-32 | 42.6 | - | - | - |
| DiVE (He et al., 2021) | ResNet-32 | 45.4 | - | - | - |
| MiSLAS (Zhong et al., 2021) | ResNet-32 | 47.0 | - | - | - |
| BS (Ren et al., 2020) | ResNet-32 | 50.8 | - | - | - |
| PaCo (Cui et al., 2021) | ResNet-32 | 52.0 | - | - | - |
| BCL (Zhu et al., 2022) | ResNet-32 | 51.9 | - | - | - |
| **Fine-tuning foundation model** | | | | | |
| Linear Prob (LA) | ViT-B/16 | 70.0 | 77.2 | 71.1 | 60.4 |
| Full-FT (LA) | ViT-B/16 | 79.6 | 88.1 | 79.9 | 69.3 |
| cRT (Kang et al., 2020) | ViT-B/16 | 78.8 | 89.7 | 79.7 | 65.1 |
| LIFT (Shi et al., 2024) | ViT-B/16 | 81.3 | 85.2 | 80.9 | 77.1 |
| RSMerge (Ours) | ViT-B/16 | **83.5** | **88.2** | **83.5** | **78.0** |

*Table 8.* Comparison of methods for training on CIFAR100-LT.

| Methods | Backbone | Overall | Many | Medium | Few |
|---|---|---|---|---|---|
| **Training from ImageNet-1K pre-trained backbone** | | | | | |
| OLTR (Liu et al., 2019) | ResNet-152 | 35.9 | 44.7 | 37.0 | 25.3 |
| cRT (Kang et al., 2020) | ResNet-152 | 36.7 | 42.0 | 37.6 | 26.4 |
| LWS (Kang et al., 2020) | ResNet-152 | 37.6 | 40.6 | 39.1 | 28.6 |
| MiSLAS (Zhong et al., 2021) | ResNet-152 | 40.4 | 39.6 | 43.3 | 36.1 |
| DisAlign (Zhang et al., 2021) | ResNet-152 | 39.3 | 40.4 | 39.4 | 32.9 |
| ALA (Zhao et al., 2022) | ResNet-152 | 41.2 | 36.1 | 47.9 | 35.3 |
| PaCo (Cui et al., 2021) | ResNet-152 | 40.5 | 33.7 | 44.4 | 35.3 |
| LiVT (Xu et al., 2023) | ViT-B/16 | 40.8 | 48.1 | 40.6 | 27.5 |
| **Fine-tuning foundation model** | | | | | |
| Linear Prob (LA) | ViT-B/16 | 48.8 | 48.8 | 49.7 | 47.1 |
| cRT (Kang et al., 2020) | ViT-B/16 | 44.4 | 51.0 | 43.1 | 35.4 |
| BALLAD (Ma et al., 2021) | ViT-B/16 | 49.5 | 49.3 | 50.2 | 48.4 |
| Decoder (Wang et al., 2023) | ViT-B/16 | 46.8 | - | - | - |
| LPT (Dong et al., 2022) | ViT-B/16 | 50.1 | 49.3 | 52.3 | 46.9 |
| Full-FT (LA) | ViT-B/16 | 46.6 | 49.9 | 46.3 | 41.4 |
| cRT (Kang et al., 2020) | ViT-B/16 | 44.4 | 51.0 | 43.1 | 35.4 |
| LIFT (Shi et al., 2024) | ViT-B/16 | **51.5** | 51.3 | 52.2 | **50.5** |
| RSMerge (Ours) | ViT-B/16 | **51.7** | 51.2 | **52.8** | 50.3 |

*Table 9.* Comparison of methods for training on Places-LT.

| Methods | Backbone | Overall | Many | Medium | Few |
|---|---|---|---|---|---|
| **Training from scratch** | | | | | |
| cRT (Kang et al., 2020) | ResNet-50 | 47.3 | 58.8 | 44.0 | 26.1 |
| LWS (Kang et al., 2020) | ResNet-50 | 47.7 | 57.1 | 45.2 | 29.3 |
| MiSLAS (Zhong et al., 2021) | ResNet-50 | 52.7 | 62.9 | 50.7 | 31.0 |
| LA (Menon et al., 2021) | ResNet-50 | 51.1 | - | - | - |
| DisAlign (Zhang et al., 2021) | ResNet-50 | 52.9 | 61.3 | 52.2 | 31.4 |
| BCL (Zhu et al., 2022) | ResNet-50 | 56.0 | - | - | - |
| PaCo (Cui et al., 2021) | ResNet-50 | 57.0 | - | - | - |
| NCL (Li et al., 2022a) | ResNet-50 | 57.4 | - | - | - |
| LiVT (Xu et al., 2023) | ViT-B/16 | 60.9 | 73.6 | 56.4 | 41.0 |
| **Fine-tuning foundation model** | | | | | |
| Linear Prob (LA) | ViT-B/16 | 74.2 | 77.8 | 73.3 | 67.4 |
| BALLAD (Ma et al., 2021) | ViT-B/16 | 75.7 | 79.1 | 74.5 | 69.8 |
| Decoder (Wang et al., 2023) | ViT-B/16 | 73.2 | - | - | - |
| Full-FT (LA) | ViT-B/16 | 73.9 | 79.8 | 71.9 | 63.9 |
| cRT (Kang et al., 2020) | ViT-B/16 | 72.6 | 81.1 | 70.6 | 56.1 |
| LIFT (Shi et al., 2024) | ViT-B/16 | 77.0 | 80.2 | **76.1** | **71.5** |
| RSMerge (Ours) | ViT-B/16 | **77.4** | **81.2** | **76.1** | 70.7 |

*Table 10.* Comparison of methods for training on ImageNet-LT.

| Methods | Backbone | Overall | Many | Medium | Few |
|---|---|---|---|---|---|
| **Training from ImageNet-1K pre-trained backbone** | | | | | |
| cRT (Kang et al., 2020) | ResNet-50 | 38.0 | 43.3 | 37.4 | 30.0 |
| LWS (Kang et al., 2020) | ResNet-50 | 28.0 | 45.7 | 23.0 | 08.3 |
| CB LDAM-DRW (Cao et al., 2019) | ResNet-50 | 37.7 | **47.6** | 35.6 | 25.0 |
| CB Softmax (Cui et al., 2019) | ResNet-50 | 33.3 | 29.5 | **41.5** | 21.7 |
| **Fine-tuning foundation model** | | | | | |
| Linear Prob (LA) | ViT-B/16 | 17.5 | 13.3 | 21.1 | 16.7 |
| BALLAD (Ma et al., 2021) | ViT-B/16 | 34.5 | 36.7 | 38.9 | 20.8 |
| Full-FT (LA) | ViT-B/16 | 38.0 | 43.8 | **41.5** | 20.0 |
| cRT (Kang et al., 2020) | ViT-B/16 | 37.7 | 42.9 | 39.3 | 25.0 |
| LIFT (Shi et al., 2024) | ViT-B/16 | 38.5 | 43.3 | 40.4 | 25.5 |
| RSMerge (Ours) | ViT-B/16 | **39.3** | 42.4 | 40.7 | **30.8** |

*Table 11.* Comparison of methods for training on NIH-CXR-LT.

| Methods | Backbone | Overall | Many | Medium | Few |
|---|---|---|---|---|---|
| **Training from scratch** | | | | | |
| cRT (Kang et al., 2020) | ResNet-50 | 65.2 | 69.0 | 66.0 | 63.2 |
| LWS (Kang et al., 2020) | ResNet-50 | 65.9 | 65.0 | 66.3 | 65.5 |
| MiSLAS (Zhong et al., 2021) | ResNet-50 | 71.6 | 73.2 | 72.4 | 70.4 |
| DiVE (He et al., 2021) | ResNet-50 | 69.1 | 70.6 | 70.0 | 67.7 |
| DisAlign (Zhang et al., 2021) | ResNet-50 | 69.5 | 69.1 | 69.9 | 69.4 |
| ALA (Zhao et al., 2022) | ResNet-50 | 69.6 | 69.5 | 70.2 | 69.0 |
| RIDE (Wang et al., 2021c) | ResNet-50 | 71.5 | 72.4 | 73.1 | 70.4 |
| RIDE+CR (Ma et al., 2023) | ResNet-50 | 73.5 | 74.0 | 74.3 | 73.1 |
| RIDE+OTmix (Gao et al., 2023) | ResNet-50 | 73.7 | 74.1 | 75.2 | 72.8 |
| BCL (Zhu et al., 2022) | ResNet-50 | 71.8 | - | - | - |
| PaCo (Cui et al., 2021) | ResNet-50 | 73.2 | 70.4 | 72.8 | 75.8 |
| NCL (Li et al., 2022a) | ResNet-50 | 74.2 | 72.0 | 74.9 | 73.8 |
| GML (Suh & Seo, 2023) | ResNet-50 | 74.5 | - | - | - |
| LiVT (Xu et al., 2023) | ViT-B/16 | 76.1 | **78.9** | 76.5 | 74.8 |
| **Fine-tuning foundation model** | | | | | |
| Linear Prob (LA) | ViT-B/16 | 60.4 | 48.9 | 60.0 | 63.9 |
| Decoder (Wang et al., 2023) | ViT-B/16 | 59.2 | - | - | - |
| LPT (Dong et al., 2022) | ViT-B/16 | 76.1 | - | - | 79.3 |
| Full-FT (LA) | ViT-B/16 | 76.1 | 75.7 | 76.9 | 75.3 |
| LIFT (Shi et al., 2024) | ViT-B/16 | **79.1** | 72.4 | **79.0** | **81.1** |
| RSMerge (Ours) | ViT-B/16 | 78.2 | 76.7 | 78.5 | 78.2 |

*Table 12.* Comparison of methods for training on iNaturalist 2018.

