# OpenReview forum: "RSMerge: Bridging Head and Tail Classes via Subsampled Model Merging"
_ICML.cc/2025/Conference — Submitted to ICML 2025_

### Official Review · Reviewer_5wyk · 2025-02-27

**Overall Recommendation:** 4

**Summary:**

This paper proposes a method called RSMerge for the long-tailed classification task which merges finetuned CLIP models on independent balanced subsets of data, then retrains the classifier on the entire dataset. A new metric, the head-to-tail ratio $\eta$ is proposed, and limitations of existing methods are identified for certain values of $\eta$. Comprehensive experiments are performed to evaluate RSMerge on a variety of challenging synthetic and real-world datasets, and good performance is identified across different values of class-imbalance ratio $\rho$ and head-to-tail ratio $\eta$.

## Update after rebuttal
The rebuttal fully addressed my concerns. This paper is strong in both *novelty*, with the introduction of the head-to-tail ratio $\rho$ and the critical assessment of LIFT, and *methodology*, with comprehensive evaluations of RSMerge in both synthetic and real-world tasks. I read through the other reviews and the additional results provided by the authors further strengthen the submission. Therefore, I recommend acceptance.

**Claims And Evidence:**

Overall, the claims made in the paper are well-supported by the presented evidence.

**Essential References Not Discussed:**

To my knowledge, essential references are sufficiently discussed.

**Experimental Designs Or Analyses:**

The experimental design, particularly with respect to the Section 4.3 empirical analyses on TinyImageNet-LT, are well-thought-out and informative. The ablation studies in Section 5.2 are also welcome.

**Methods And Evaluation Criteria:**

The methodology and evaluation criteria is a strength of the paper. The “tricks of the trade” employed to address degradation of pretrained features are clearly explained in Section 4.1. Multiple evaluation metrics (overall accuracy, many/medium/few-shot accuracy) are utilized to understand fine-grained performance of the proposed method. Comparison against state-of-the-art methods is comprehensive. Finally, both synthetic tasks (such as the proposed CIFAR-100-LT which enables manipulation of $\eta$) and multiple challenging real-world datasets are considered.

**Other Comments Or Suggestions:**

Below, I’ve included a list of typos or sentences where more clarification is needed.

1. Usage of periods after \paragraph is not consistent.
2. Line 112/114: \cite should be used instead of Author et al (Author et al 2024).
3. Line 163: Quotation mark is backwards.
4. Line 189: \ll should be used instead of <<.
5. Line 199: Extra space after (Figure 5).
6. Line 291: “Classification” typo.
7. Line 322: Malformed sentence: “Table 1 An”.

**Other Strengths And Weaknesses:**

A strength of the paper is its introduction of a new metric, the head-to-tail ratio $\eta$, which complements the commonly used class-imbalance ratio $\rho$. The finding that LIFT compromises head-class accuracy for certain values of $\eta$ is novel and interesting, and the proposed CIFAR-100-LT benchmark used to investigate $\eta$ is well-thought-out.

A minor weakness is that the computational tradeoffs of RSMerge against previous work are not made explicit. Specifically, the “computational analysis” subsection of Section 4.3 should include a more detailed comparison between RSMerge and competing methods including LIFT and full finetuning. Currently, the full number of training runs is listed along with the subsample sizes and a brief comment on memory. The subsection would be improved with a table describing, e.g., the wall-clock training time, GPU VRAM usage, and disk usage at each stage of RSMerge.

A compute statement is missing, e.g., concerning the hardware used to run the experiments and/or total cost in cloud compute credits.

**Questions For Authors:**

Which hyperparameters are tuned on the validation set? In the paper, only temperature (for the calibration study) and early-stopping duration are mentioned. More specifically, how were $\lambda, M, N$ chosen?

**Relation To Broader Scientific Literature:**

This paper identifies a shortcoming of previous algorithms utilizing PEFT for long-tailed classification: specifically, that LIFT [2] may compromise head-class accuracy. The proposed method, RSMerge, draws from related literature including model merging [3] and vision-language foundation models [1].

[1] Radford et al. Learning transferable visual models from natural language supervision. ICML 2021.

[2] Shi et al. Long-tail learning with foundation model: Heavy fine-tuning hurts. ICML 2024.

[3] Tarvainen and Valpola. Mean teachers are better role models: Weight-averaged consistency targets improve semi-supervised deep learning results. NeurIPS 2017.

**Theoretical Claims:**

No theoretical claims are made.

---

> ### Author Rebuttal · Authors · 2025-04-01
>
> We thank the reviewer for the thoughtful feedback and positive comments on our novel findings, rigorous methodology, and thorough empirical analyses. We address the questions as follows:
>
> ---
> > Q1. I would encourage the authors to include a README explaining how to install/run the code.
>
> We want to highlight that we have included a run.sh script for reproducibility and will add a detailed README with step-by-step instructions upon the official code release.
>
> ---
> > Q2. Computational analysis.
>
> Thanks for the suggestion. The table below compares the computational costs of Full Fine-Tuning (Full-FT), LIFT (which employs a LoRA adapter with rank 64 applied to all MLP layers), and RSMerge on the ImageNet-LT and CXR-LT datasets. All models were trained to convergence using a batch size of 128 and mixed-precision training. For RSMerge, we break down the computational cost into two stages. Stage 1 involves training models independently and in parallel on subsets with different imbalance ratios. Since the subset with the highest imbalance ratio contains the most training samples, it dominates the overall wall-clock time. Stage 2 retrains only the linear classifier on the full dataset—a highly efficient step, as it updates just a single linear layer. In our experiments, we used the same number of epochs for both stages of RSMerge.
>
>
> | Method      | Wall-clock Time | Training Iterations | Param (M) | Mem (G) | Acc  |
> |-------------|-----------------|---------------------|-----------|---------|------|
> |||**ImageNet-LT**||
> | Full-FT               | 1:37:56        | 9060                | 87.0      | 14.5    | 73.9 |
> | LIFT           | 1:25:33        | 9060                | 9.0       | 13.3    | 77.0 |
> | RSMerge Stage 1 (Rep Learning)       | 1:15:38        | 8050                | 87.0      | 14.5    | 76.7 |
> | RSMerge Stage 2 (Classifer Re-Train)  | 0:30:0        | 9060                | 0.7       | 2.6     | 77.4 |
> | Full RSMerge  | 1:48:38        |                 |        |      | 77.4 |
> |||**CXR-LT**||
> | Full-FT               | 0:53:43        | 5320                | 87.0      | 14.5    | 38.0 |
> | LIFT           | 2:14:32        | 13300               | 9.0       | 13.3    | 38.5 |
> | RSMerge Stage 1 (Rep Learning)       | 0:12:51        | 1300                | 87.0      | 14.5    | 37.8 |
> | RSMerge Stage 2 (Classifer Re-Train)  | 0:19:26        | 5320                | 0.7       | 2.6     | 39.3 |
> | Full RSMerge  | 0:32:17        |                 |        |      | 39.3 |
>
> The computational overhead of RSMerge compared to existing methods depends heavily on dataset characteristics—particularly the original imbalance ratio. For example, in ImageNet-LT, which has an imbalance ratio of 256, the largest subset used in Stage 1 accounts for 89% of the full training data, resulting in relatively higher wall-clock time. In contrast, on CXR-LT, with a much more extreme imbalance ratio of 6401, the largest Stage 1 subset represents only 24% of the dataset, leading to a 4.4× reduction in training time compared to Full-FT (see Table 7 in the appendix for details). Additionally, while full-rank methods like Full-FT and RSMerge typically converge within 10 epochs on CXR-LT, LIFT required 50 epochs—substantially increasing its wall-clock time despite its parameter-efficient design.
>
> ---
> > Q3. A compute statement is missing, e.g., concerning the hardware used to run the experiments and/or total cost in cloud compute credits.
>
> All experiments were conducted on a cluster equipped with NVIDIA RTX 3090 GPUs (24GB VRAM), using Python 3.9.15, PyTorch 2.4.0, and CUDA 11.8.
>
> ---
> > Q4. Which hyperparameters are tuned on the validation set? In the paper, only temperature (for the calibration study) and early-stopping duration are mentioned. More specifically, how were $\lambda$, $M$, and $N$ chosen?
>
> Across all datasets, we fix $M=2$. As discussed in Section 5.2, we validate two $\lambda$ values (0.3 and 0.7): datasets closely aligned with CLIP's pre-trained distribution benefit from a higher $\lambda$, while others perform better with a lower value. For $N$, we use imbalance factors that are powers of 2, ensuring the maximum factor is less than half of the full dataset's imbalance (see Table 7). We use the same number of epochs as in LIFT—except for CXR-LT, where hyperparameter search spans all baselines. Finally, we use an equal number of epochs for both RSMerge stages.
>
> ---
> > Q5. Other Comments Or Suggestions.
>
> Per the reviewers’ request, we have implemented the suggested changes, including ensuring consistent usage of periods after \paragraph, correcting citation formatting (L112/114), fixing the backward quotation mark (L163), replacing << with \ll (L189), removing the extra space after "(Figure 5)" (L199), correcting the typo in "Classification" (L291), and fixing the malformed sentence in L322 ("Table 1 An").

---

> > ### Comment · Reviewer_5wyk · 2025-04-02
> >
> > Thank you for the detailed response; my concerns are fully addressed. This paper is strong in both _novelty_, with the introduction of the head-to-tail ratio $\rho$ and the critical assessment of LIFT, and _methodology_, with comprehensive evaluations of RSMerge in both synthetic and real-world tasks. I read through the other reviews and the additional results provided by the authors further strengthen the submission. Therefore, I recommend acceptance.

---

> > > ### Author Response · Authors · 2025-04-09
> > >
> > > Dear Reviewer 5wyk,
> > >
> > > Thank you for your positive response! We extend our appreciation once again for your recognition of our work!
> > >
> > > Best,
> > >
> > > Authors.

---

### Official Review · Reviewer_jM36 · 2025-03-02

**Overall Recommendation:** 3

**Summary:**

This paper explores class imbalance recognition, identifies head-to-tail class ratio as an under-explored problem in this setup, and proposes an approach that successfully integrates the main challenges in imbalanced class learning. The proposed approach leverages observations from the literature to navigate the trade-off between maintaining the model information from a pre-training representation learning stage and integrating new information from a calibration or classifier-training stage that addresses the imbalance in the data. The authors propose to train several models in parallel on subsets of the data as a pre-training/representation learning stage, and then merge and freeze the weights of these models to train a classifier with all the dataset.

**Claims And Evidence:**

Overall, the claims are supported by convincing evidence. However, I would suggest the authors clarify where the improvements come from (first or second stage of the proposed approach) and empirically show the benefits of the different design decisions. In particular, it would be valuable to see if the weight averaging of the models provides performance improvements regardless of the subsets' design (as would happen with traditional ensemble methods): the sizes of the subsets, imbalance ratios, etc.

**Essential References Not Discussed:**

NA

**Experimental Designs Or Analyses:**

The experiments section is comprehensive and well designed to support the main claims in the text. However, the simplicity of the method proposed might come across as a weakness. To address this, I would suggest further analysis of the different stages of the approach to illustrate its contributions. For instance, it would be valuable to include the performance of the models from the initial stage when trained on the different subsets with different imbalance factors or an analysis of the contribution of the two stages separately. This would help the reader see the contribution of the different elements of the approach.

**Methods And Evaluation Criteria:**

Yes, the experimental section is well designed to show the effectiveness of the proposed approach to learn from imbalanced datasets.

**Other Comments Or Suggestions:**

- The titles of the x-axis in figures 1 (top) and 2 look a bit untidy. I would suggest centering them with respect to the full figure.
- Figure 2: the reader would benefit if the stages (1) and (2) were highlighted or indicated in the figure, not only in the caption.
- Fig 2 Caption: finish with a period.
- L294: typo, repeated words.

**Other Strengths And Weaknesses:**

Strengths
- The authors successfully identify an underexplored challenge in imbalanced training (head-to-tail class ratio).
- Tthe paper is well written and easy to follow.
- The method proposed is simple and effective with competitive results in most of the experiments across several datasets.

Weaknesses
- As a minor comment, the paper would benefit from some visualization or experiment showing how the proposed approach bridges head and tail classes (as stated in the title). This is currently observed by comparing the performance of "many", "medium", "few" in the results tables. However, this is not linked to a particular element from the approach. I would suggest including a discussion or visualization showing how the approach "bridges head and tail classes".

**Questions For Authors:**

NA

**Relation To Broader Scientific Literature:**

The authors integrate several observations from the literature to address the biases learned during training in imbalanced datasets. This leads to valuable insights.

In particular, the authors successfully integrate and examine the effect of existing techniques to fine-tune foundational models. This is complimented with discussions and observations that help explain the applicability of existing approaches (eg LoRa, LIFT).

The second stage of the approach leverages LA, an existing technique to learn under class imbalance. While this is successfully included in the approach, it would be valuable to further expand on the role of this technique in conjunction with the other elements of the aproach.

**Theoretical Claims:**

NA

---

> ### Author Rebuttal · Authors · 2025-04-01
>
> We thank the reviewer for their insightful feedback and for recognizing the value of our work on imbalanced training. Our responses to their questions, which we found fruitful and led to several new experiments and analyses, are below:
>
> ---
> > Q1. It would be valuable to see if the weight averaging of the models provides performance improvements regardless of the subsets' design (as would happen with traditional ensemble methods): the sizes of the subsets, imbalance ratios, etc.
>
> Thanks for the suggestion. For TinyImageNet-LT, which has an original imbalance ratio of 100, RSMerge averages the weights of 14 models: 7 models are trained on subsets with progressively increasing imbalance ratios (from 1 to 64, doubling at each step), with 2 models per subset obtained via resampling at the same ratio. To demonstrate the impact of our proposed weight averaging scheme in RSMerge, we compare this with a WA-{imbalance ratio} baseline that averages 14 models, each trained with a fixed imbalance. Notably, WA-100 aligns with the popular model soup approach (Wortsman et al., 2022a), where the weights of 14 fully fine-tuned models on the entire dataset are averaged.
>
> As noted by the reviewer, weight averaging consistently outperforms full fine-tuning, regardless of the subset design. However, results show that different imbalance ratios yield varying outcomes across head and tail categories. For example, WA-{8} achieves the highest tail accuracy of 75.0, whereas WA-{Full} reaches the highest head accuracy of 85.9. Rather than optimizing for a single imbalance ratio, RSMerge applies weight averaging across the full spectrum, effectively merging the advantages of both approaches to achieve a more balanced overall trade-off.
>
> |     | RSMerge | WA-1 | WA-2 | WA-4 | WA-8 | WA-16 | WA-32 | WA-64 | WA-100 (Full) |
> | --- | --- | --- | --- | --- | --- | --- | --- | --- | --- |
> | Acc | 78.3 | 74.6 | 75.9 | 76.0 | 77.2 | 77.2 | 77.3 | 77.9 | 77.6 |
> | Head | 84.6 | 76.8 | 78.6 | 78.7 | 81.0 | 82.8 | 84.7 | 85.5 | 85.9 |
> | Tail | 75.0 | 72.9 | 74.4 | 74.6 | 75.0 | 74.1 | 73.3 | 73.7 | 73.0 |
>
> ---
> > Q2. Effect of LA loss.
>
> The table below demonstrates the impact of replacing LA loss with either a cross-entropy (CE) or a class-balanced sampling (CB) approach. Unlike LIFT, which relies heavily on LA loss for optimal performance, RSMerge is only partially sensitive to the choice of loss function—thanks to its inherent balance achieved through subsampling and weight averaging.
>
> | Method | Acc | Head | Tail |
> | --- | --- | --- | --- |
> ||**CE**||
> | LIFT | 72.6 | 85.1 | 65.9 |
> | RSMerge | 76.3 | 84.5 | 71.9 |
> ||**CB**||
> | LIFT | 75.3 | 81.0 | 72.2 |
> | RSMerge | 78.2 | 84.5 | 78.4 |
> ||**LA**||
> | LIFT | 77.1 | 83.0 | 73.9 |
> | RSMerge | 78.3 | 84.6 | 75.0 |
>
> ---
> > Q3. An analysis of the contribution of the two stages separately. This would help the reader see the contribution of the different elements of the approach.
>
> Figure 4 of the paper breaks down the contribution of each stage in the RSMerge pipeline. To further clarify these contributions, we present the contents of the Figure in a simplified tabular format below. In particular, the introduction of EMA enhances performance for both head and tail classes while slightly reducing weight magnitude. Progressive subsampling, which averages multiple models trained on less imbalanced distributions, effectively limits weight changes and boosts tail-class accuracy—albeit at the cost of reduced head-class performance. Resampling and classifier re-training then restore head-class accuracy by averaging models trained on a fixed imbalance ratio and fine-tuning the classifier on the full dataset, respectively.
> Notably, RSMerge achieves a slightly higher weight magnitude than LIFT, which we argue contributes to its superior final performance by striking an optimal balance between adaptation and stability.
>
> | Method | Acc | Head | Tail | Weight Change Magnitude |
> | --- | --- | --- | --- | --- |
> | Full-FT | 73.2 | 83.4 | 67.7 | 35.4 |
> | + EMA | 74.1 | 84.0 | 68.8 | 35.1 |
> | + Prog. Subsampling | 77.9 | 83.1 | 75.1 | 12.7 |
> | + Resampling | 78.1 | 84.5 | 74.7 | 11.9 |
> | + Classifer Re-Train (RSMerge) | 78.3 | 84.6 | 75.0 | 12.1 |
> | LIFT | 77.1 | 83.0 | 73.9 | 10.3 |
>
> ---
> > Q4. It would be valuable to include the performance of the models from the initial stage when trained on the different subsets with different imbalance factors.
>
> Please refer to the answer to Q1.
>
> ---
> > Q5. How does RSMerge "bridge head and tail classes"?
>
> Due to space constraints, we refer you to rev. EBKi rebuttal response for “Q2. Why the proposed method can solve this problem”.
>
> ---
> > Q6. Other Comments Or Suggestions.
>
> Per the reviewers’ request, we have implemented the suggested changes, including centering the titles in Figure 1, correcting the typo in L294, updating the caption in Figure 2, and highlighting the two stages of RSMerge in Figure 2.

---

### Official Review · Reviewer_82gt · 2025-03-10

**Overall Recommendation:** 2

**Summary:**

The paper proposes a method for long-tailed recognition by using CLIP. The proposed method trains multiple models from different distributions and then merges them together. While training each model, the exponential moving average is applied to maintain the original generalizability of pre-trained CLIP.

**Claims And Evidence:**

- The authors' claim on head-to-tail imbalance is sound.

- Please cite proper reference for the statement that “LoRA enhances tail-class performance by maintaining weight close to the pre-trained initialization, yet it sacrifices head-class accuracy in Section 4.

**Essential References Not Discussed:**

Many papers related to long-tailed recognition are ignored, especially the loss function perspective (e.g., Equalization Loss v2, Balanced Softmax). I recommend that authors write a section for general long-tailed recognition not limited to architectural approaches. Maybe the authors might want to put in the Appendix.


- Tan, Jingru, et al. "Equalization loss v2: A new gradient balance approach for long-tailed object detection." Proceedings of the IEEE/CVF conference on computer vision and pattern recognition. 2021.

- Ren, Jiawei, et al. "Balanced meta-softmax for long-tailed visual recognition." Advances in neural information processing systems 33 (2020): 4175-4186.

**Experimental Designs Or Analyses:**

- The authors evaluated their methods on the standard long-tailed recognition benchmark with the standard metrics.

- There are various hyperparameters such as $\lambda$ and $N$, $M$, and $\rho$. The authors need to show how the model is sensitive to those parameters.

**Methods And Evaluation Criteria:**

- The authors' claim on head-to-tail imbalance is sound.

- Please cite proper reference for the statement that “LoRA enhances tail-class performance by maintaining weight close to the pre-trained initialization, yet it sacrifices head-class accuracy in Section 4.

**Other Comments Or Suggestions:**

- The citation format is wrong. Please carefully differentiate \citet and \citep. For example, L112 Alexandrov et al.,\citep{} should be just \citet{}. And L121 (right column) \citep{radford} also \citet{}.

- Please unify the format in L425 left and L411 right. (\lambda$) and $N$.

**Other Strengths And Weaknesses:**

I do not have any further strength and weakness comments.

**Questions For Authors:**

- I do not have any further questions.

**[post rebuttal]**
Several concerns have been resolved. I recommend that authors add the results of various $\lambda$ in the appendix (at least 0.3, 0.5, 0.7). I have no objection to accepting this paper.

**Relation To Broader Scientific Literature:**

I do not have a concern regarding the broader impact.

**Theoretical Claims:**

There is no theoretical proof or analysis in this paper.

---

> ### Author Rebuttal · Authors · 2025-04-01
>
> We thank the reviewer for their feedback and address their concerns in detail below:
>
> ---
> > Q1. Please cite proper reference for the statement that “LoRA enhances tail-class performance by maintaining weight close to the pre-trained initialization, yet it sacrifices head-class accuracy in Section 4.
>
> This is one of our key findings, one that has not been observed in previous LT literature. As demonstrated and empirically verified in Sections 3.2 and 4.3 (weight magnitude analysis) of the paper, this phenomenon is novel within the long-tail context. Notably, at the end of Section 3.2 (L204), we also observe that our findings align with recent trends in the LLM literature, where low-rank approximation methods such as LoRA often underperform full fine-tuning while better preserving the base model's performance on tasks outside the target domain (Biderman et al., 2024).
>
> ---
> > Q2. There are various hyperparameters such as \lambda, M, N, and \rho. The authors need to show how the model is sensitive to those parameters.
>
> First, note that $\rho$ is not a hyperparameter; it is a dataset-dependent variable that quantifies the proportion of head-to-tail classes (see Def. 3.1). Additionally, Section 5.2 of the paper outlines the rationale behind our choice of hyperparameters. To further support our design decisions, we conduct an ablation study on TinyImageNet-LT, which has an original imbalance ratio of 100, examining:
>
> - $N$ represents the number of subsampling steps, ranging from 1 corresponding to the perfectly balanced dataset, up to $\log$(imbalance ratio). By convention, we double the imbalance ratio at each step, though alternative curriculums are possible.
> - $M$ denotes the number of models per subset obtained via resampling at the same ratio.
> - $\lambda$: the factor controlling the preservation of previously merged knowledge through progressive subsampling.
>
> The table below summarizes our findings. As the subset imbalance ratio increases, head-class performance consistently improves improves—from 74.5 at $N=1$ to 84.8 at $N=8$. In contrast, the highest tail-class performance of 76.1 is achived at $N=4$. The best overall trade-off between head and tail performance occurs at $N=7$, indicating a balanced configuration. A similar trend is observed for weighting paramter $\lambda$: a higher value emphasizes earlier subsets with lower imbalance ratios, benefiting tail classes, while lower value shifts focus toward head-class performance. Lastly, resampling helps recover information lost due to subsampling, as reflected in the performance gain from 78.1 to 78.3.
>
>
> | $\lambda$ | $M$ | $N$ |  Acc  | Head | Tail |
> |  --- | --- | --- | --- | --- |--- |
> | 0.7 | 2 | 7 | 78.3 | 84.6 | 75.0 |
> | 0.7 | 1 | 7 | 78.1 | 84.5 | 74.7 |
> | 0.3 | 2 | 7 | 77.2 | 84.8 | 73.1 |
> | 0.7 | 2 | 8 | 78.2 | 84.8 | 74.6 |
> | 0.7 | 2 | 6 | 77.9 | 83.1 | 75.0 |
> | 0.7 | 2 | 5 | 77.7 | 81.9 | 75.4 |
> | 0.7 | 2 | 4 | 77.9 | 81.2 | 76.1 |
> | 0.7 | 2 | 3 | 75.9 | 78.5 | 74.5 |
> | 0.7 | 2 | 2 | 74.0 | 76.5 | 72.6 |
> | 0.7 | 2 | 1 | 71.5 | 74.5 | 69.9 |
>
> ---
> > Q3. Many papers related to long-tailed recognition are ignored, especially the loss function perspective (e.g., Equalization Loss v2, Balanced Softmax).
>
> As noted in L107, the claim made by the reviewer that we ignored “Balanced Softmax” is incorrect: it is well known in the long-tail community that Balanced Softmax is a special case of the logit adjustment loss function, with the temperature fixed at 1.
>
> ---
> > Q.4 I recommend that authors write a section for general long-tailed recognition not limited to architectural approaches.
>
> There is an extensive body of literature on long-tail and imbalanced recognition approaches. However, due to space constraints, we focus on the methods most closely related to our work in the main text; in response to the reviewer's request, we have expanded the literature review in the appendix.
>
> ---
> > Q5. Other Comments Or Suggestions.
>
> We have corrected the citation formatting errors and typos.

---

> > ### Comment · Reviewer_82gt · 2025-04-04
> >
> > I appreciate the authors' response. I will respond it as early as possible.
> >
> >  Before then, I realized the Method Section was not copied when I posted a review. I apologize for the inconvenience.
> >
> > Below is my original comment regarding the `Methods And Evaluation Criteria` Section. Although I will not degrade anything because of the concerns I had earlier in this section, I would like to share them to improve the paper in the future.
> >
> > ```
> > Although each model learns a portion of the entire dataset, the authors need to train MN models. This might require a much longer time of training compared to other methods. Please measure the training time compared to other methods. Considering this overhead, the performance improvement is marginal or even slightly worse than one of the baseline methods (LiFT).
> >
> >
> > How to determine $\lambda$ in Eq. 5
> > ```
> >
> > **Response to Authors**
> >
> > > Regarding Q1.
> >
> > Thank you for pointing out the relevant section. However, the authors did not compare it with vanilla LoRA but used LIFT, where the final model uses AdaptFormer, another PEFT. Moreover, Table 6 in Shi et al. (2024) already shows that LoRA enhances tail class compared to full fine-tuning. Hence, the authors claim that `LoRA enhances tail-class performance, yet it sacrifices head-class accuracy` is their finding might not be true. Furthermore, it is a well-known property that LoRA maintains weight close to the pre-trained initialization compared to full fine-tuning (e.g., Biderman et al., 2024; Hu et al., 2021). I do not think the authors found that `LoRA enhances tail-class performance by maintaining weight close to the pre-trained initialization, yet it sacrifices head-class accuracy. ` If so, the authors theoretically prove this property.
> >
> > > Regarding Q2.
> >
> > Although $\rho$ is measured by dataset (L192), $\rho$ for each expert is arbitrary determined (L248). Section 5.2 does not explain how to choose but explains empirical outcomes (L427-431, L412-418 right). The sensitivity analysis for N in the rebuttal is helpful,l but I still cannot get rationals to use only 2 modules (L262), although the framework can be expanded to more modules. Moreover, authors mentioned that they empirically observed the outcomes of different $lambda$, but since there is no experiment, readers find it difficult to accept the claim.
> >
> > > Regarding Q3, Q4.
> >
> > Thank you for pointing this out. However, I still believe that equalization loss v2 and other methods should be used to complete related works. If space is constrained, it might be okay to put it into the appendix.

---

> > > ### Author Response · Authors · 2025-04-07
> > >
> > > We appreciate the reviewer’s insightful feedback, which has inspired several new experiments and analyses. Below, we address their questions in detail:
> > >
> > > > ## Computational Analysis.
> > >
> > > Due to space constraints, we refer you to Rev. 5wyk rebuttal response for “Q2. Complete computational analysis”.
> > >
> > > > ## In section 3.2, the authors did not compare it with vanilla LoRA but used LIFT, where the final model uses AdaptFormer, another PEFT.
> > >
> > > We observed the same trend with both AdaptFormer and LoRA, as both optimize low-rank adaptations of the full weight matrix (see https://imgur.com/a/qyIVa9G for the updated Fig.2 using both methods). This clarification has been added to the text.
> > >
> > > > ## Table 6 in Shi et al. (2024) already shows that LoRA enhances tail class compared to full fine-tuning. Hence, the authors claim that LoRA enhances tail-class performance, yet it sacrifices head-class accuracy is their finding might not be true.
> > >
> > > We acknowledge that the observation—“LoRA enhances tail-class performance, yet it sacrifices head-class accuracy”—can be deduced from Tab. 6 in the LIFT paper and may seem obvious in hindsight. However, as also acknowledged by the Rev 5wyk and jM36, we underscore this trade-off by critically assessing LIFT across a range of head-to-tail ratios, $\eta$, in both synthetic (CIFAR100-LT) and diverse real-world datasets (CXR-LT, iNaturalist). These insights have motivated the development of our method, which achieves a more balanced performance between head and tail classes.
> > >
> > > > ## LoRA maintains weight close to the pre-trained initialization compared to full fine-tuning.
> > >
> > > We agree that LoRA’s ability to keep weights closer to their pre-trained initialization, as opposed to full fine-tuning, is well-established; we have already noted a relevant example in the LLM community (L204). However, we believe our key observation lies in highlighting how this property relates to long-tail and imbalanced recognition tasks—specifically, the trade-off between head and tail performance. While providing a formal proof is beyond the scope of this work, we empirically verify this phenomenon in Section 4 and leverage it to develop our proposed method. We would be happy to include any additional references if the reviewers feel we have overlooked important work in this area.
> > >
> > >
> > > > ## Outcomes of different $\lambda$
> > >
> > > We reiterate that $\lambda$ controls the preservation of merged knowledge via progressive subsampling. The table below compares $\lambda = 0.3$ and $\lambda = 0.7$ on four datasets—CXR-LT, iNaturalist, ImageNet, and Places-LT—each split into many-shot (>100), medium-shot (20–100), and few-shot (<20) subsets. As shown in Section 5.2, datasets similar to CLIP’s pre-training distribution (ImageNet and Places-LT) benefit from a higher $\lambda$, while CXR-LT and iNaturalist perform better with a lower value. This selection is further supported by analyzing the performance gap between ***linear probing*** and ***Full-FT*** baselines on each dataset (Tab.3,4,5,6): a large gap indicates that the pre-trained representations are already well-suited for the downstream task, reducing the need for extensive feature adaptation, whereas a smaller gap suggests that additional adaptation is required for optimal performance.
> > >
> > >
> > > |||$\lambda$=0.3|||||$\lambda$=0.7|||
> > > |-|-|-|-|-|-|-|-|-|-|
> > > ||Overall|Many|Med|Few|-|Overall|Many|Med|Few|
> > > |**CXR-LT**|**39.3**|**42.4**|**40.7**|**30.8**|-|37.8|42.9|41.5|20.8|
> > > |**iNaturalist**|**78.2**|**76.7**|**78.5**|**78.2**|-|77.7|73.7|78.2|78.0|
> > > |**ImageNet-LT**| 76.2|81.2|74.6|67.8|-|**77.4**|**81.2**|**76.1**|**70.7**|
> > > |**Places-LT**  | 51.2|52.2|52.2|48.8|-|**51.7**|**51.2**|**52.8**|**50.3**|
> > >
> > >
> > > > ## imbalance ratio $\rho$ for each expert is arbitrary determined.
> > >
> > > Please note that knowing $N$ (the number of subsampling steps) and the functional form for increasing the subset size (in our case, $2^N$) is enough to determine each expert's $\rho$. While alternative growth functions are an interesting future direction, our core message remains: varying imbalance ratios yield different head and tail outcomes. Rather than tuning a single imbalance ratio, RSMerge averages weights across an increasing range, merging the benefits of diverse strategies for a more balanced trade-off (see Q2, Rev EBKi).
> > >
> > > > ## Rationale to use M as 2.
> > >
> > > $M$ is the number of models per subset, with each subset resampled at the same imbalance ratio ($\rho$). The table below shows that increasing $M$ on TinyImageNet-LT yields higher overall, head, and tail accuracy. To keep experiments manageable across five datasets, we fix $M = 2$, although we could obtain a higher performance in the sota tables with larger $M$.
> > >
> > > ||M=1|M=2|M=6|M=12|
> > > |-|-|-|-|-
> > > |Acc|78.1|78.3|78.6|78.8|
> > > |Head|84.5|84.6|85.0|85.5|
> > > |Tail|74.7|75.0|75.2|75.2|
> > >
> > > > ## I still believe that equalization loss v2 and other methods should be used to complete related works.
> > >
> > > Thanks for your suggestion. We have expanded the literature review in the appendix.

---

### Official Review · Reviewer_EBKi · 2025-03-13

**Overall Recommendation:** 2

**Summary:**

This paper conducts a comprehensive analysis of head-to-tail class ratios under different levels of class imbalance, investigating their effects on model performance. Building on these findings, this paper proposes a two-stage approach to address the stability-plasticity dilemma through decoupled learning and model merging, balancing the accuracy of head and tail classes across various conditions. Experiments on five datasets demonstrate the effectiveness and generalizability of the proposed method in real-world applications.

**Claims And Evidence:**

The author claims that existing methods often fail to adapt to varying head-to-tail class ratios. In other words, across different head-to-tail ratios, the accuracy of the dominant class is often higher than that of the tail class. Through the analysis of Section 4, ‘RSMerge: Imbalanced Learning by Controlling Weight Change’, as well as the experimental results presented in Figure 1 and Tables 2, 3, 4, 5, and 6, this paper demonstrates that the proposed method effectively reduces the accuracy gap between head and tail classes across different head-to-tail ratios.

**Essential References Not Discussed:**

None

**Experimental Designs Or Analyses:**

The proposed method is evaluated on five datasets, demonstrating its advanced performance. Furthermore, detailed ablation experiments validate the effectiveness of the proposed modules. This paper provides a comprehensive analysis of both comparative and ablation experiments.

**Methods And Evaluation Criteria:**

The proposed method is indeed reasonable and effective. Classification accuracy is used to evaluate the model's performance.

**Other Comments Or Suggestions:**

No other comments or suggestions

**Other Strengths And Weaknesses:**

Strengths

1.The problem addressed in this paper is stated clearly, and a considerable portion of the paper is dedicated to describing imbalanced learning and the proposed method in an accessible manner.

2.Extensive experiments are conducted on five datasets to validate the robustness of the proposed method.


Weaknesses

1.The abstract presents an excessively detailed background but lacks a comprehensive description of the proposed method.

2.Although the paper provides a detailed explanation of the reasons behind the problem it aims to address, it does not thoroughly elaborate on why the proposed method can solve this problem. Instead, it devotes a substantial portion of the text to explaining how the method is implemented.

3.Figure 3 lacks clarity and needs further refinement. It is recommended to present the overall training process to enhance the understanding of RSMerge.

4.What concerns me is that the paper does not include an analysis of the algorithm's complexity, such as FLOPs, GPU memory usage, or training time. These experimental data should be added to the paper for further clarification.

5.The paper lacks a discussion of the algorithm’s implementation details, which limits the method’s transparency and reproducibility.

6.The performance improvement of the proposed method is relatively limited, with only a 0.2% increase on the Places-LT dataset, a 0.4% increase on the ImageNet-LT dataset, and a 0.9% decline on the iNaturalist 2018 dataset.

**Questions For Authors:**

None

**Relation To Broader Scientific Literature:**

The proposed method builds upon previous approaches, such as LIFT [1], which applies parameter-efficient fine-tuning to CLIP's visual encoder.
[1] Long-tail learning with foundation model: Heavy fine-tuning hurts.

**Theoretical Claims:**

I carefully examine the theoretical claims in Section 4, ‘RSMerge: Imbalanced Learning by Controlling Weight Change’, and find them logically coherent. This section sequentially introduces Progressive Subsampling, Progressive Resampling, and Classifier Re-Training, followed by an empirical analysis of RSMerge.

---

> ### Author Rebuttal · Authors · 2025-04-01
>
> We appreciate the reviewer's careful reading, positive feedback on our method's accessibility and effectiveness, and constructive questions that spurred additional experiments. Below, we address their questions in detail:
>
> ---
> > Q1. Lack of a comprehensive description of the proposed method in the abstract.
>
> We will follow your suggestion and improve the description of our method in the abstract.
>
> ---
> > Q2. Why the proposed method can solve this problem.
>
> **Motivation:**
> Section 3.2 reveals that head and tail performances are inherently opposed, driven not only by the imbalance ratio but also by our introduced head-tail ratio. Most LT methods boost tail accuracy at the cost of head-class performance. For instance, on CIFAR100-LT with a head-tail ratio of 19 (95 head classes) (Fig. 2), LIFT improves tail results via low-rank adaptation but underfits head classes, resulting in an overall performance of 84.9 versus 86.3 for Full-FT. This trade-off suggests that while full-rank adaptation excels at capturing head-class information, it often sacrifices tail performance by straying from the pre-trained configuration.
>
> **Our Objective:**
> We aim to sustain robust performance across both head and tail classes, irrespective of the head-tail scenario—a challenge that has been largely unexplored in LT literature. Our goal is twofold: we want the flexibility of full-rank optimization to effectively capture head-class information, and we encourage balanced learning between head and tail classes to prevent the model from being biased toward head classes.
>
> **Philosophy of RSMerge:**
> RSMerge merges full-rank optimization with balanced learning via progressive subsampling. While subsampling improves tail-class performance (Chaudhuri et al., 2023), it loses head-class data, harming head performance. We address this by training independent models on subsets with increasing imbalance ratios. Each model specializes in a segment of the imbalance spectrum, and their average produces a final model robust across the range. We substantiate this insight with a detailed ablation study below.
>
> **Empirical evidence:**
> RSMerge averages the weights of $N \times M$ models, where $N$ is the number of subsampling steps (from 1 for a perfectly balanced dataset up to $\log$(imbalance ratio), doubling the ratio at each step) and $M$ is the number of models per subset via resampling. For TinyImageNet-LT (imbalance ratio 100), we use $N=7$ and $M=2$. The table below compares RSMerge with WA-{imbalance ratio} baselines—each averaging 14 models trained with a fixed imbalance ratio. Notably, WA-100 corresponds to the popular model soup approach (Wortsman et al., 2022a).
>
>
> |     | RSMerge | WA-1 | WA-2 | WA-4 | WA-8 | WA-16 | WA-32 | WA-64 | WA-100 (Full) | Full-FT |
> | --- | --- | --- | --- | --- | --- | --- | --- | --- | --- | --- |
> | Acc | **78.3** | 74.6 | 75.9 | 76.0 | 77.2 | 77.2 | 77.3 | 77.9 | 77.6 | 73.2 |
> | Head | 84.6 | 76.8 | 78.6 | 78.7 | 81.0 | 82.8 | 84.7 | 85.5 | **85.9** | 83.4 |
> | Tail | 75.0 | 72.9 | 74.4 | 74.6 | **75.0** | 74.1 | 73.3 | 73.7 | 73.0 | 67.7 |
>
> Results show that different imbalance ratios yield varying outcomes across head and tail categories. For example, WA-{8} achieves the highest tail accuracy of 75.0, whereas WA-{100} reaches the highest head accuracy of 85.9. Rather than optimizing for a single imbalance ratio, RSMerge applies weight averaging across the full spectrum, effectively merging the advantages of both approaches to achieve a more balanced overall trade-off.
>
> ---
> > Q3. Refinement of Figure 3.
>
> Figure 3 illustrates the overall training steps involved in RSMerge. To enhance clarity, we explicitly highlight Stages 1 and 2 of the pipeline in the figure.
>
> ---
> > Q4. Computational analysis.
>
> For space reasons, we refer you to rev. 5wyk rebuttal response for “Q2. Computational analysis”.
>
> ---
> > Q5. Lack of algorithm’s implementation details.
>
> We describe our algorithm in Section 4, and detail hyperparameters, experimental setup, and additional information in Section 5.2 and Appendices A and B. The source code is also available. Please let us know which implementation details are unclear or hinder reproducibility.
>
> ---
> > Q6. The performance improvement of the proposed method is relatively limited.
>
> Addressing class imbalance requires considering the head-to-tail ratio—a critical but often overlooked factor. While traditional benchmarks focus on tail-dominated scenarios, real-world datasets like CXR-LT have a majority of head classes. In such cases, enhancing tail performance without sacrificing head accuracy is essential. Our method, RSMerge, achieves this by combining full-parameter fine-tuning for head-class information with weight averaging across progressively subsampled subsets, resulting in a better trade-off (RSMerge = 39.3 vs. LIFT = 38.5).

---

### Decision · Program_Chairs · 2025-05-01

**Decision:**

Reject

**Comment:**

After reviewing the paper, it received two positive and two negative reviews. Following the authors' rebuttal, all reviewers maintained their original positions. During the AC-reviewer discussion, only one reviewer participated—the reviewer who initially gave a rating of 2 noted that several concerns had been addressed in the rebuttal (though they did not update their rating).

Given this, the AC conducted a thorough assessment of the paper and identified several key limitations in its current version:

- The paper argues that head-to-tail class ratios are the primary factor influencing performance. However, since different datasets exhibit varying long-tail label distributions, this ratio is just one statistical aspect of the overall distribution. Other factors—such as the max/min number of training samples per class and the total training sample size—could also significantly impact model performance. The authors should provide stronger justification that the head-to-tail ratio is indeed the most critical factor.

- The technique contribution of the paper is limited. First of all, the decoupling training strategy builds directly on prior work [1]. Second, while the paper claims that the proposed method avoids distorting pre-trained representations (a point closely related to [2]), it lacks rigorous theoretical or empirical proof. Third, the techniques employed (EMA, model averaging, PEFT) are well-established and do not represent novel contributions.

- The AC also reviewed the Author AC Confidential Comments. Although not all reviewers specialize in long-tail learning, most of their critiques—including questions about the method’s rationale and marginal improvements over LIFT—are valid.

Given these issues, the AC recommends rejection. However, the authors are encouraged to carefully revise the paper, addressing the reviewers’ feedback to strengthen both its technical contributions and presentation.

[1] Decoupling Representation and Classifier for Long-Tailed Recognition. ICLR 2021.

[2] Fine-Tuning can Distort Pretrained Features and Underperform Out-of-Distribution. ICLR 2022.